# ZnO and ZnO-Based Materials as Active Layer in Resistive Random-Access Memory (RRAM)

**Ewelina Nowak *** , **Edyta Chłopocka and Mirosław Szybowicz ***

Insitute of Materials Research and Quantum Engineering, Poznań University of Technology, 61-139 Poznań, Poland
* Correspondence: ewelina.nowak@put.poznan.pl (E.N.); miroslaw.szybowicz@put.poznan.pl (M.S.)

**Abstract:** In this paper, an overview of the influence of various modifications on ZnO-based RRAM has been conducted. Firstly, the motivation for creating new memory technology is presented. The resistive switching mechanism is explained, including its response to the selection of active layers and electrodes. A comparison of ZnO devices assembled via different deposition methods is made. Additional treatment of the active layer and electrodes improving the performance are reported. This work gives an overview of the influence of different dopants on the characteristics of the device. The manuscript overviews the previous investigation of inclusion of inserting layers and nanostructures into ZnO-based RRAM.

**Keywords:** ZnO; $MoS_2$; graphene; doping; memristor; resistive switching; RRAM

## 1. Introduction

The last decades in human history can be called "the big data era". Contemporary applications such as artificial intelligence, cloud storage, data mining, or the internet of things were possible due to the advances in data storage technology. Modern applications require high velocity and generate a large volume of data with less energy consumption. The conventional von Neumann architecture with silicon complementary metal-oxide-semiconductor systems (CMOS) and charge-based memory makes power scaling easier, as the charge leaks away easily in a smaller device. Therefore, non-charge-based memory technologies such as resistive random access memory (RRAM) have become promising for future applications [1,2].

Today, for data operation, both temporary and permanent storage are required. Currently, these demands are fulfilled with dynamic random access memory (DRAM), static random access memory (SRAM), and Flash memory [2]. A DRAM cell uses a capacitor to store charge and distinguish between the '0' state and the '1' state. The cell scaling is narrowed by the load of charge, which is accumulated in the scaled capacitor [2]. An SRAM cell stores information on the two nodes of a cross-coupled inverter pair. It is a very fast memory used to interact directly with the high-speed processor. However, an SRAM is volatile and has a lower density. Flash memories are employed for large capacity and nonvolatile requirements. A flash memory cell stores charge in the floating gate of a transistor and can store different amounts of charge to effectively store more than one bit of information per transistor [2]. All these existing charge-storage-based memory technologies face challenges in scaling down to 10 nm nodes or beyond. This is correlated with stored charge loss at the nanoscale, which results in the decrease in performance and reliability, performance, and widening of the noise margin. Furthermore, the leakage power for both SRAM and DRAM and requirements of high dynamical refresh power for DRAM pose serious design challenges [3].

Resistive random access memory (RRAM) devices have appeared as a potential candidate for the forthcoming flexible non-volatile memory (NVM) device due to their distinctive features such as scalability, higher speed operation, CMOS compatibility, and low power

consumption. Therefore, in broad investigations have been carried out on RRAMs, focusing on improving their performance and eliminating limitations such as the high impact of process-induced variations [2]. Moreover, RRAM may be used in neuromorphic systems as synapse emulators [4], where one of the challenges is the lack of a compact analog RRAM that bridges the gap between the fundamental physics of the device and the behavior of the circuit/system [5].

RRAM, also often referred to as a memristor, is a non-volatile memory made from the simple structure of a metal–insulator–metal (MIM) sandwich, which is generally integrated into an elementary crossbar circuit [6,7]. Memristor is the physical realization of the fourth fundamental passive circuit element [8]. Its primary role is resistive switching. The device relies on the formation of conducting filaments to switch between low- and high-resistance states. This property makes it particularly useful for in-memory computing due to its non-volatile storage capability with a continuum of conductance states [9,10]. RRAM uses an electrical signal to activate the reversible transition between a high resistance state (HRS, OFF) and a low resistance state (LRS, ON) in a sandwiched structure, thus enabling the storage of data '0' and '1' [8,11,12]. Due to the characterization of the materials, the most crucial feature of MIM switches is their HRS/LRS switching ratio (the higher the ratio, the better the memristive behavior) [13].

Despite the theory of MIM switching having been introduced more than 50 years ago, the explanation of the driving mechanism was only presented two decades later. The phenomenon of negative differential resistance in oxides was first reported in the 1960s and reviewed by Dearnaley et al. in 1970 [12]. Researchers then saw reversible resistive switching in various binary oxides [14]. In 1971, Chua discovered the memristor as the fourth fundamental electrical element [7]. The latest research on resistive switching can be set on the discovery of I-V hysteresis in perovskites and binary metal oxides in the late 1990s and early 2000 [12,14]. These findings began enormous interest in resistive switching in oxides for application in RRAMs. Research activity intensified after 2004 when Samsung presented an article demonstrating NiO memory cells integrated with conventional CMOS in a one-transistor–one-resistor (1T1R) architecture [15]. Hewlett-Packard Labs achieved the first clear connection between Chua's theory and practical demonstrations of memristor devices in 2008. The group observed a memristive behavior at the nanoscale using thin film titanium dioxide as an insulator layer [6,16].

Today, more and more research is focused not only on the usage of oxides, organic materials, or 2D nanostructures as an insulating layer, but also on the modification of the properties of each insulating layer by adding other materials that can change the properties of the film. Since one of the most commonly utilized materials in memristors is ZnO, this paper focuses on understanding a review of RRAM based on ZnO thin films modified with 2D and 1D materials.

## 2. RRAM Mechanism

As mentioned above, a random access memory resistor (RRAM) consists of a memory cell of resistance switching with a metal–insulator–metal structure, generally known as the MIM structure. The structure consists of a layer (I) of insulation between two metal electrodes (M) [3]. The number of electric charges flowing through it can reversibly modulate the memristor's resistive states. The memristive device performs resistive switching behavior with an inherent memory effect. The resistive state depends both on the extra stimulations and its intrinsic states [17].

Depending on different criteria, the behavior of resistance exchange can be classified into different types. For example, resistive switching behavior can be divided into digital and analog categories based on switching dynamics. Digital resistance switches describe sudden changes between high resistance states (HRS) and low resistance states (LRS), and sudden current jumps appear in the digital cell I–V loops. Analog switching relates gradual modulation—switching cells exhibit continuous I-V loops [17]. According to the retention

characteristics of the resistive states, the resistive switching behavior can be classified into volatile and non-volatile switching groups [17].

The switch from HRS to LRS is the 'set' process. In contrast, the LRS–HRS switch event is called a 'reset' process. Applying the external voltage pulse through the RRAM cell allows a transition of the device from a high resistance state (HRS), or the OFF state generally referred to as logic value '0', to a low resistance state (LRS), or the ON state—logic value '1' and vice versa. The resistance change phenomenon (RS) is considered the reason behind the change in resistance values in RRAM cells. In most cases, in new samples with initial resistance conditions, a voltage greater than the set voltage is required to trigger resistive switching behaviors in subsequent cycles [3,15]. To read data from the RRAM cell, a small voltage which cannot destroy the current state is applied for determination whether the cell is in the logic 0 (HRS) or logic 1 (LRS) state. Since LRS and HRS maintain their respective values even after applying voltage, RRAM is a non-volatile memory [3].

### 2.1. Resistive Switching

According to the current polarity, the RRAM can be divided into two modes: unipolar and bipolar (Figure 1). In unipolar switching, changing between modes does not depend on the polarity of the applied voltage. In bipolar switching, the SET and RESET processes rely on the polarity of the applied voltage. A switch from an HRS to an LRS occurs at one polarity (positive or negative) and the opposite polarity shifts the RRAM cell back into the HRS [3]. Resistance-switching properties in memristive devices were highly dependent on materials, device structures, external simulations, and switching mechanisms. Therefore, many reports deal with the adjustment of switching behavior characteristics by modifying the structure of the device and the simulation parameters [11,15].

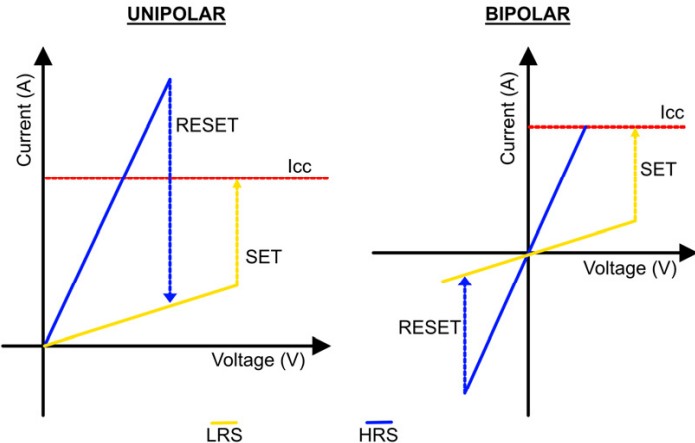

**Figure 1.** Unipolar and bipolar modes for RRAM devices.

The basis of a switching mechanism is the growth of a conductive filament (CF) inside the insulator. A CF is a very narrow channel that connects the top and bottom electrodes of the memory cell. Low-resistant (LRS) states with high conductivity are achieved when the filament is connected, and high-resistance state (HRS) is obtained when the filament is disconnected from the gap between the electrodes. Based on the composition of the conductive filament and the mechanism of conduction, RRAM switching can be classified as a thermal chemical mechanism (TCM), valance change mechanism (VCM), and electrochemical metallization (ECM) [3,11].

Electrochemical metallization (ECM) is based on the migration of metal ions and subsequent reduction/oxidation (redox) reactions. The junction consists of a chemically active top electrode such as Ni, Cu or Ag, a nearly inert bottom electrode (e.g., W, Pt), and a sandwiched metal oxide layer. The filament formation in such memory cells occurs due to the dissolution of the active metal electrodes, the transport of cations ($M^+$), and their deposition or reduction on the inert electrode [3,11].

In the valence change mechanism, the formation of a conduction filament is correlated with the creation of oxygen vacancies ($V^{2+}_O$) and the subsequent relocation of oxygen ions ($O_2$), thus enabling the formation of a conductive filament between the upper and lower electrodes of RRAM cell. For the commissioning of the mechanism, it is necessary to knock out oxygen atoms from the lattice by applying a high electric field toward the anode interface. The oxygen ions ($O_2$) drift toward the electrode whereas the oxygen vacancies ($V^{2+}_O$) are left in the oxide layer. If noble metals are used as materials for the anode to form an interfacial oxide layer, oxygen ions ($O^{2-}$) react with anode materials or are released as neutral oxygen. Next, the conductive filament (CF) is formed and the appreciable current flows in the device through the accumulation of oxygen vacancies ($V^{2+}_O$) in the bulk oxide, which switch the RRAM cell to the low resistance state (LRS). To return the device to the high resistance state (HRS), the reset process occurs. In the process, the oxygen ions ($O_2$) migrate back to bulk oxide from the anode interface and combine with the oxygen vacancies ($V^{2+}_O$) [3,11].

The thermochemical mechanism (TCM) explains the formation and fracture of CFs resulting from ion migration induced by a thermochemical reaction (Joule heating), which is independent of the switching modes (unipolar and bipolar) for RRAM devices. In the case of LRS, the ions are driven by the Joule heating effect towards the top electrode and, in the case of the unipolar device, left oxygen vacancies. For the RESET process of the unipolar device, the current steadily increases with increasing positive voltage bias, and the formed CFs finally break when it reaches the critical temperature induced by Joule heating, which causes the device to switch back to HRS. For the RESET process of the bipolar device, oxygen ions drift back to the insulating layer due to the melting of CF and the device to HRS [11].

### 2.2. Activation Process

The RRAM behavior is based on the possibility of electrically modifying the conductivity of a stack of metal–insulator–metal (MIM). To activate the switching mechanism, some technologies require a preliminary formation operation [18], which is shown in Figure 2. The electroforming process (soft or hard) is usually realized by applying a large electrical bias across the two electrodes within a certain specific time interval to generate initial conductive channels via the Joule heating effect. The formation step can be conquered by appropriately modifying the fabrication process to readily introduce oxygen vacancies to facilitate the migration of anions within the switching layer [6]. Even if the forming process is performed once, this initial state plays a fundamental role in determining the subsequent array and system performance. The performance of the formation process relies on its ability to create homogeneous conductive conditions among cells, thus easing successive SET/RESET operations [18]. As explanations of the driving force of anion transport during the formation process, the following are suggested: (i) drift by electric potential gradient, (ii) electromigration assuming an electron kinetic energy, (iii) Fick diffusion due to ion concentration gradient, and (iv) thermophoresis due to temperature gradient [6].

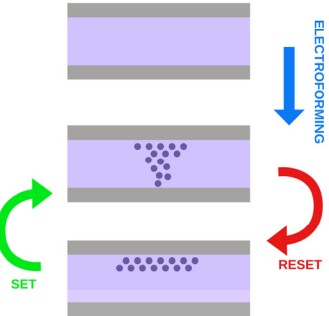

**Figure 2.** Process of electroforming, SET and RESET.

A standard formation can be carried out by applying a voltage ramp [19] or a voltage/current pulse to each cell individually [18]. Both formation processes produce a non-destructive soft breakdown regime and a progressive breakdown regime of the dielectric and require a sufficiently high electric field [20]. Another method, the constant voltage formation process, enables the formation of conductive filaments at lower voltages rather than the conventional fast voltage ramp method [21].

### 2.3. Material for Electrodes

One of the undervalued elements of MIM switches is the electrode. The materials and forms of electrodes can have a significant impact on the behavior of RS, mainly through direct participation in redox reactions or as transport routes for oxygen tanks or loading carriers [9].

Firstly, it is necessary to choose a suitable material for the top and bottom electrodes. In general, the specific material for the electrodes is related to the conduction mechanism.

VCM is triggered by the migration of field-assisted oxygen anions and valence change of the cation sublattice [22]. In the case of VCM, the most common and widely used are inert metals, such as Au, Pt, and Pd, which normally contribute less to RS, and act mainly as a carrier transport path or oxygen reservoir [9].

In the case of ECM, the conduction mechanism is based on the migration of cations in the solid electrolyte [b]. The typical ECM cell has an asymmetric structure with one active electrode. In the case of cation-based structures, electrochemically active metals such as Cu, Ni, Ag, and Ru, have been explored [9]. Active metals with the ability to modulate the concentration or migration of anions are used, such as Ta, Ti, Al, Hf, and W. What is more, the redox reaction of a Ta electrode can lead to the formation of Ta CFs. Therefore, competition between the ECM and the VCM may exist in devices based on the Ta electrode [9]. Recently, active metals have been intentionally mixed into the electrolyte layer to achieve bipolar threshold-switching behavior with typical I–V curves. These memristive devices with bipolar threshold-changing behaviors are also called diffusive memristive devices [17].

More and more alloy electrodes are used for improving or optimizing RS behaviors by modulation of the mobile cations' diffusivity or confinement of the position of CFs [9]. The use of alloys where both components are redox-active and mobile may bring significant advantages in the design of electrode materials [23]. Most commonly used switching films are binary, ternary, or quaternary compounds such as $Ta_2O_5$, $SiO_2$, $SrTiO_3$, etc. In those alloys, the interactions between the ions/atoms of the active electrode/filament and the solid electrolyte matrix influence the chemical and electronic properties both of the conducting channel and also of the whole matrix [23].

Carbon-based electrodes, for example, graphene and carbon nanotubes (CNT), are reported for flexible and small-scale devices. One of the more common bottom electrodes is p- and n-type silicon, as well as nitride electrode materials such as TiN and TaN. Conducting metal oxides, such as indium tin oxide (ITO), Al-doped ZnO (AZO), Ga-doped ZnO (GZO), and F-doped $SnO_2$ (FTO), have also been reported as electrodes for some special applications, for example, fully transparent or flexible devices [9].

The electrode material can significantly change the behavior of the active layer. Khrapovitskaya et al. [24] have investigated RS of $TiO_2$-based memristors with respect to different material of top electrodes. In the case of the Pt electrode, the maximum to minimum resistance ratio ($R_{off}/R_{on}$) was up to about 100 $\Omega$, whereas in the case of the golden electrode, the sample resistance in the low-ohmic state ($R_{on}$) was about 90 $\Omega$, and that in the high-ohmic state ($R_{off}$) was about 900 $\Omega$. The author claimed that the greater switching behavior in the case of memristors with gold electrodes is probably related to a lower diffusion of oxygen through the gold film compared to the case of platinum. On the other hand, Kumar et al. [25] have investigated the effect of different electrode materials in the case of the active ZnO layer of ZnO. They observed that the ZnO-based memristor with the Pt electrode showed a better hysteresis compared to Cr and Au metal electrodes. In the

case of the Pt electrode, a current ratio of six times the magnitude was observed between the high resistive state and the low resistive state at 1 V, where a maximum current density value of 1.25 A/cm$^2$ was measured.

Swathi et al. [26] investigated the changes in RS due to the bottom electrode. They investigated the Au/NiO/ITO and Au/NiO/Pt devices. Although the NiO switching layer was deposited under similar conditions, different switching patterns were observed in NiO films with ITO and Pt electrodes. In particular, the RS device with ITO as BE exhibits gradual set-and-reset switching or analog-type RS. The change from HRS to LRS and vice versa in positive and negative voltage sweeps was incremental in comparision to abrupt change in samples with Pt bottom electrode device [26].

In addition to material, the size of the top electrode influences the formation of the memristor conduction structure [27]. Gale et al. [28] showed that the scaling of the electrode changing affects the behavior of curved-type memristors and has no effect on triangular-switching ones. This suggests that the two types operate via different mechanisms. The value of the hysteresis increases with increasing electrode size as a result of the decrease in the value of R$_{on}$ with increasing electrode size [28].

In another work, Gale et al. [29] observed a larger I-V curve in the case of a larger electrode, which indicates the memristor's response under the electrical field in relation to three spatial dimensions. Furthermore, the hysteresis increases with electrode size but does not increase equally across the devices; instead, the top right quadrant of the curve increases more. This asymmetry leads to a negative hysteresis [29].

The size and materials of the electrodes may present some problems. Schroeder et al. [30] observed the molting between the top and bottom electrodes as a result of a large current during the forming process. Similarly, the fusing of electrodes lying next to each other was observed, or the observation of creating the dendritic structures on the substrates [30].

### 2.4. Material for Active Layer

Many thin film materials have been investigated as RS mediums for RRAM devices because of their RS characteristics under the influence of the external electrical field. Generally, organic materials and inorganic materials are two categories of RS medium [11].

### 2.4.1. Organic Materials

In the case of organic materials, research focuses primarily on biological materials, polymer materials, and other materials. Most of them require low-temperature processes. Therefore, in most cases, developing a way to control their interfacial pathways is a milestone [31]. For example, investigation on the memory effect began in the 1970s when the switching mechanism between different resistance states was observed in polystyrene and copper–tetracyanoquinodimethane (Cu–TNCQ) films. Since then, noteworthy progress has been made in organic memory devices [32].

Polymers are the most common group of organic materials used as active layers. One of the most commonly used materials is polyaniline (PANI). The researchers utilized its adaptive behavior: the PANI demonstrates non-linear electrical characteristics with hysteresis loop and rectification [33,34]. Berzina et al. [35] used the PANI difference in the conductivity in the oxidized and reduced states for the memristive behavior. PANI may also act as an electrode, and was utilized by Erokhin et al. [36] in a sandwich structure, where a solid electrolyte (polyethylene oxide doped with lithium salt (PEO)) acted as an active layer.

More and more researchers are focusing on producing flexible devices. For example, Xu et al. used a chlorotrifluoroethylene and vinylidene fluoride copolymer (FK-800) to produce RRAM for an artificial nociceptor (pain sensor) [37]. Another realization of a flexible memristor was proposed by Zhou [38], who used polymer nanocomposites, with the configuration of the silver nanowire (AgNW)/citric acid quantum dot (CA QD)-polyvinyl pyrrolidone (PVP)/AgNW. Park et al. [39] proposed poly(vinyl cinnamate) (PVCi) with a predefined CF.

Many polymer devices are based on complexes with 2D material or composites. One of the mediums most commonly chosen for different materials is PMMA [31,40,41] or azobenzene polymer [42].

Due to its rich electrochemical redox behavior, viologen diperchlorate EV(ClO4) with different polymers (pyridyl-iron polymer (TPy-Fe) [43], triphenylamine-containing polymer (BTPA-F) [44] are used to simulate the functions of the synapse. Other materials used in the production of memristors are (PVK (polyvinyl carbazole), PVA (polyvinyl alcohol), PDA (polydiacetylene), and PTH (polythiophene) [11].

Another widely used group of organic substances are biomaterials. Biomaterial-based memristive devices are made of biopolymers produced by organisms. These substances can be divided into two groups: carbohydrates and protein. The molecular structure of carbohydrates contains only three elements: carbon (C), hydrogen (H), and oxygen (O). However, in addition to C, H, and O, proteins usually contain nitrogen (N) from amino acids and some trace elements such as iron (Fe), zinc (Zn), copper (Cu), manganese (Mn), and so on. Thus, the memristive effect depends on trace elements, which help to form conductive filaments and redox reactions [32]. Natural organic materials can provide versatile engineering platforms and are an attractive alternative due to their biodegradability, bioabsorbability, and nontoxicity [45].

One of the most commonly used biomaterials is egg albumen [11,45]. The albumen layer is characterized by a transparency of more than 90% under visible light with a wavelength range of 230–850 nm, flexibility [32,46], clockwise and counterclockwise current hysteresis [47]. Low SET/RESET voltage ~3 V and reliable switching endurance were observed over 500 cycles with ~$10^3$ ON/OFF ratio [11].

Spider silk [48] and silkworm cocoon fibroins [11,49,50] are highly utilized in the production of MIM junctions. The fibroin structures exhibit excellent performance and behave as RRAM with the immersion process in di-isopropanol water, indicating the great potential of silk fibroin applied to transient and biocompatible electronics [11].

Battistoni et al. reported two hybrid devices based on poly(3,4-ethylenedioxy- thiophene), doped with polystyrene-sulfonate (PEDOT: PSS) and Physarum polycephalum slime mold (PP), which acted as a living electrolyte [33]. Furthermore, Abbas et al. fabricated and characterized the transparent and biocompatible resistive random access memory (ReRAM) device with the structure of Pt/$Cu^{2+}$ doped salmon DNA/FTO where $Cu^{2+}$ was doped in salmon DNA by solution processes [32].

Protein-based memristors were produced mainly with keratin [32] and gelatin [51]. However, polysaccharides are also widely used in the manufacture of memristors. The MIM structures are based on chitosan [32,45], cellulose [11], or glucose [11]. Furthermore, resistive switching behavior was also found in orange peel pectin [32], maple leaves [52], and anthocyanin extracted from plant tissue [53].

Organic small molecules that have a clear structure, easy purification, and low cost, and are lightweight also have caused a widespread boom in research in the field of resistive switch memory. Furthermore, in the production of RRAM, liquid crystalline polymer (LCP) aligned with polyimide was used [54]. Copper phthalocyanine nanowires were used to change the response due to IR illumination [55].

### 2.4.2. Inorganic Materials

Compared to organic RRAM, inorganic materials exhibit better electrical performance, more stable switching behavior, lower energy consumption, and longer retention time [11]. Hickmott proposed the first report on RS performance in binary metal oxides in 1962, which demonstrated the RS characteristics of the Al/$Al_2O_3$/Al device under the effect of an electric field [11]. Inorganic memristors usually have a typical metal–insulator–metal (MIM) structure, and their insulator layer (also known as the RS layer) is made of binary metal oxides, perovskite metal oxides, chalcogenides, and others [56].

The layer of binary metal oxides is usually formed with a single insulator such as $TiO_2$ [11,57], NiO [11], $HfO_2$ [11,58], $SiO_2$ [59], $TaO_2$ [11,60], and $Ga_2O_3$ [61]. However,

with the development of research, researchers tend to make a dielectric layer diversification by doping or making a multilayer. Two or three dielectric layers have various degrees of optimization effects on the performance of the device [56]. Sakellaropoulos et al. demonstrated a comparison among other devices with three types of dielectric structures such as $HfO_x$, $TaO_y/HfO_x$, and $HfO_x/TaO_y/HfO_x$, which correspond to single-layer (SL), bilayer (BL), and triple-layer (TL) [11]. Liu et al. [56] investigated conductance modulation on TE/$HfO_x$/$AlO_x$/BE and Ta/$TaO_x$/$TiO_2$/Ti stacks. Mahata et al. reported an RRAM device with ALD-based $HfO_2$/$Al_2O_3$ stack layers, which exhibited excellent performance with an operating voltage lower than ~2 V and an ON/OFF ratio [11]. Prezioso et al. investigated spike time-dependent plasticity on 200-nm $Al_2O_3$/$TiO_{2-x}$ memristors integrated into $12 \times 12$ crossbars [62].

For a memristive performance, porous bionic structures are investigated. One of the best examples is the work of Gao et al. [63], who researched a double-layer structure comprising a Pt/porous $LiCoO_2$/porous $SiO_2$/Si stack. Furthermore, binary oxides are mixed with nanomaterials such as nanotubes, which may increase reservoir oxygen vacancies [11].

Another group of materials that are used is perovskites. Perovskite is a compound with the $ABX_3$ type crystal. A is a monovalent cation and can be an organic (methylammonium $CH_3NH_3^+$) or inorganic ($Cs^+$) cation. B is a divalent cation, such as $Pb^{2+}$, and $Sn^{2+}$, and X is an anion. When X is oxygen (O), the material is called an oxide perovskite, and when X is a halide (I or Br), it is called an HP [64]. The oxide perovskite, which was investigated, had higher dielectric constants, and those are $LaAlO_3$, $SrTiO_3$, $Pr_{0.7}Ca_{0.3}MnO_3$, and $BiFeO_3$ [11]. Halide perovskite (HP) materials with point defects (such as gaps, vacancies, and inversions) also have a strong application potential in memristors with an averaged ON/OFF ratio of $10^4$–$10^5$ [64,65]. Unfortunately, most HPs are lead-based: $MAPbI_3$, $FAPbI_3$, $HC(NH_2)_2PbI_3$, $CsPbI_3$, or $(Cs_3Bi_2I_9)_x$- $(CsPbI_3)_{1-x}$. Thus, the thermal instability and toxicity severely restricted their further practical applications. Therefore, more and more researchers are focusing on lead-free HP such as $CsSnI_3$, $Cs_3Bi_2I_9$, $(MA)_3Bi_2I_9$, $(BzA)_2CuBr_4$, or $CsBi_3I_{10}$ [66,67].

Organic–inorganic halide perovskites (OHPs) have gained attention as promising materials for memristors. Particulary, their mixed ionic-electronic conduction ability paired with light sensitivity allows OHPs to show novel functions such as optical erase memory, optogenetics-inspired synaptic functions, and light-accelerated learning capability [68]. Furthermore, to enhance the properties of perovskites, the material is mixed with nanostructures, for example, reduced graphene oxide (rGO) [69]. In addition, amorphous perovskite materials exhibit memristive properties [70].

A series of 2D materials such as graphene and molybdenum disulfide (disulfide perovskite) have gained popularity due to their small size, ultrathinness, and excellent physical properties, which have resulted in the performance of RRAM devices [11]. For example, additional graphene layers can act as the charge storage medium, resulting in a higher retention time. In addition, graphene can offer increased transparency, light weight, flexibility, and low sheet resistance [13]. Nonvolatile and bistable memory devices based on graphene oxide (GO) have prompted great interest due to their high optical transparency, low cost, easy fabrication, high flexibility, environmentally friendly nature, and controllable chemical and physical properties for future electronic devices [13]. On the other hand, $MoS_2$ embedded in the active layer tends to trap and release charge carriers [13].

## 3. ZnO as Active Layer

The resistive switching effect is demonstrated by many metal oxides [71–73], among which zinc oxide is an especially prominent material due to its broad band gap, chemical stability, high thermal conductivity, and melting point [74]. Additionally, zinc oxide has been widely used for its biodegradable and biocompatible properties. That is why ZnO-based RRAM can not only have excellent operational characteristics [75] but also find application in the field of green electronic devices and implantable biomedical devices [76,77]. ZnO, in the last two decades, has also gained interest as a potential material for flexible and

transparent electronics, e.g., transistors and solar cells [78]. Flexible ZnO-based RRAM has been successfully assembled and does not change its properties significantly after bending tests [79–81].

ZnO thin films for RRAM applications have been produced with various techniques, mainly techniques requiring vacuum conditions such as sputtering [75,77,81–90], pulsed laser deposition [91,92], and plasma-enhanced atomic layer deposition [93]. A significant amount of research is also focused on alternative methods of obtaining devices such as inkjet printers [81] and chemical routes such as electrochemical deposition [76], sol-gel [79,80,94,95], or chemical bath deposition [96]. Worth mentioning is that the memory performance of sol–gel ZnO films prepared by a spin-coating, which is a significantly easier and cheaper technique, is comparable to that of ZnO films prepared using conventional vacuum deposition processes [79,94].

The annealing process has a tremendous influence on ZnO layer properties because it is one of the factors influencing crystallinity [94]. As shown in Hsu et. al. [97], crystallinity of the active layer is a key factor in the device performance. The conduction mechanism in polycrystalline ZnO is usually correlated with the hopping of electrons through filament paths consisting of oxygen vacancies. The set process is induced by a defect-induced soft breakdown, which is associated with a polarization effect due to the migration of oxygen vacancies under an applied electric field [82]. Oxygen vacancies are formed and accumulated around the grain boundaries so the conducting filaments follow the grain boundaries and make conductive paths. As there are fewer grain boundaries that can potentially become conductive paths in films with a higher degree of crystallinity, the current paths can be reduced in the HRS as the annealing temperature increases [79]. Amorphous ZnO may present resistive switching performance as good as the polycrystalline one [98] but be in turn attributed to the formation of the electrode's metal nanofilaments in amorphous ZnO. One of the techniques to improve the switching cycle is $N_2$ rapid thermal annealing (RTA) of films. Presumably, the larger and condensed grains with grain boundary paths are beneficial to the HRS/LRS resistance switching and improve the switching cycle [83]. High temperature annealing is responsible for weakening the screening effect which accounts for the absence of resistive switching characteristics in devices. Annealing can effectively reduce the number of defects and the carrier concentration in ZnO films, thus increasing the driving force of oxygen vacancy drifting and improving the device performance. A study researching the influence of annealing temperature on the screening effect [88] compared thin films of ZnO fired in 300, 450 and 600 °C. In conclusion, the higher temperature of annealing allows for more effective devices. The observation was confirmed later by Gupta [99].

Sputtering is commonly used for assembling ZnO-based RRAM devices; therefore, the influence of various sputtering factors on RRAM devices properties has been extensively researched. Reactive magnetron sputtering films vary in properties depending on oxygen gas flow ratio during growth. As the oxygen–gas flow ratio increases, Zn atoms tend to bind to oxygen atoms in the ZnO film. Specifically, when more oxygen atoms are incorporated into the ZnO film, the binding energy decreases slightly [9]. Another important factor is working pressure, which can impact devices' power consumption and ensure stable high-endurance properties [87]. Radio frequency magnetron sputtering can give tremendously different device characteristics, especially crystallinity level, depending on substrate temperature during deposition [97].

As mentioned earlier, the conduction mechanism and its effectiveness depend on the electrodes' material. For example, RRAM devices with TiN top electrode maintain good switching endurance. Ti and TiN are the scavenging metals and are used to attract oxygen from RS layers for oxygen vacancies creation [93]. TiN electrode act as an oxygen reservoir and provide sufficient non-lattice oxygen ions to recover the oxygen vacancies during the reset process [82]. In another study Pt, Au, and Cr electrodes were compared [100]. The ZnO-based RRAM devices with Pt electrodes had the lowest activation energy for the oxygen chemisorption process, which leads to better switching functionality due to

the increased generation/elimination probability of oxygen vacancies. Nevertheless, DFT calculations [101] show that two Pt electrodes are not compatible when the ZnO active layer consists of nanowires. Comparison between Ag, Ti, and Pt electrodes showed that different top electrodes can make a difference in the switching mechanism, and the devices with a metallic conductive bridge mechanism have more prominent switching behavior than those with an oxygen ion/vacancy filament mechanism [88]. ZnO biocompatibility can be fully explored when paired with graphene electrode, as in Tian et. al. [77].

Another factor impacting the conduction mechanism is interface morphology. For example, one of the electrodes, instead of flat thin film, can consist of arrays of periodic nanotips. As has been reported in Tsai et. al., the electric field concentrated on nanotip structures plays a crucial role in lowering $V_f$ and $V_{set}$. Similarly, RRAM with a rough surface of the top electrode has shown superior switching probability and stable characteristics against various conditions [86]. The mechanism responsible for the phenomena is believed to be the roughness-enhanced absorption on oxide surface. When the switching mechanism is a metallic conductive bridge, a rough surface is also superior due to enhanced diffusion of ions into the resistive layer, which leads to easier filament formation, as explained in Wu et. al. [95]. Additional UV–ozone treatment of the Pt electrode may provide superior surface morphology for the subsequent sputtering ZnO thin film deposition [102].

Table 1 presents a selection of some of the pure ZnO-based RRAM devices produced in the last 15 years. The highest HRS/LRS ratio was achieved via vacuum techniques such as sputtering (up to 1010). Plasma-enhanced atomic layer deposition allowed for a 105 switching ratio. Sol-gel films' RRAM HRS/LRS ratio varies from $10^2$ [94] to $10^6$ [79,80]. Electrochemical deposition technique is not very commonly utilized despite the HRS/LRS ratio reaching $10^3$. A low but sufficient for device operation ratio of 5–40 was documented for PLD films [91,92]. Similarly, CBD method allowed for RRAM with 18.7 ratio [96].

As can be seen, ZnO thin films have been extensively researched for RRAM devices with a variety of results. Even within one deposition technique, performance can be significantly changed depending on parameters such as thickness, electrode material, additional treatment, etc.

**Table 1.** Comparison between selected RRAM devices, ND—no data.

| Device (Thickness) | ZnO Layer Deposition Method | Additional Information | Ref. | $V_{RESET}$ [V] | $V_{SET}$ [V] | HRS/LRS Ratio |
|---|---|---|---|---|---|---|
| Al/ZnO (30 nm)/Al | sol-gel | | [79] | 3.0 | 0.8 | >$10^4$ |
| Al/ZnO (37 nm)/Al | | Si substrate | [80] | 0.6–0.8 | 1.5–1.8 | >$10^2$ |
| | | flexible substrate | | 0.3–0.6 | 1.5–1.8 | >$10^4$ |
| Al/ZnO (20 nm)/Cu | | changing Cu electrode roughness | [95] | ND | ND | $7.7 \times 10^4$–$3.1 \times 10^6$ |
| Al/ZnO/Al | | | [94] | ~0.6 | ~1.7 | >$10^2$ |
| GZO/ZnO (90 nm)/GZO | PLD | | [91] | 1.5–1.8 | 2.0–2.4 | 5–10 |
| Ti/ZnO/Pt | | | [92] | −2.5 | 4 | ~41 |
| ITO/ZnO/Ag | CBD | nanorod layer | [96] | −2.78 | 3.25 | 18.7 |
| Ag/ZnO (100 nm)/W | electrochemical deposition | | [76] | −2.8 | 3.1 | $10^3$ |
| Au/ZnO/AZO | plasma-enhanced atomic layer deposition | | [93] | ND | ND | $10^5$ |
| EDOT:PSS/ZnO/PEDOT:PSS | jet-printing | | [81] | −3.5 | 0.7 | 5 |
| Pt/ZnO (20 nm)/TiN | sputtering | | [89] | −0.7 | 0.7 | >$10^2$ |
| TiN/ZnO (30 nm)/Pt | | | [82] | −4.0 | 4.0 | ~10 |
| Al/ZnO (60nm)/Al | | no rapid thermal annealing | [103] | 0.6 | 2.2 | $10^8$ |

**Table 1.** *Cont.*

| Device (Thickness) | ZnO Layer Deposition Method | Additional Information | Ref. | $V_{RESET}$ [V] | $V_{SET}$ [V] | HRS/LRS Ratio |
|---|---|---|---|---|---|---|
| TiN/ZnO (30 nm)/Pt | | rapid thermal annealing | [82] | 0.3 | 2.6 | $10^9$ |
| Al/ZnO (71.4 nm)/Al Al/ZnO (70 nm)/Al | | | [75,85] | 2.5 | 0.5 | $10^9$ |
| | | oxygen-gas flow ratio 16% | | 2.85 | 0.3 | $\sim 10^5$ |
| Al/ZnO (71.4 nm)/Al | | oxygen-gas flow ratio 25% | [75] | 2.45 | 0.35 | $\sim 10^9$ |
| Al/ZnO (70 nm)/Al Pt/Cr/SiO$_2$/Si/ZnO (100 nm)/Pt | sputtering | oxygen-gas flow ratio 33% | [85,86] | 2.30 | 0.25 | $\sim 10^8$ |
| | | rough interface | | 1.5 | 1.65 | $\sim 717$ |
| | | flat interface | | 2.0 | 2.6 | $\sim 4600$ |
| Ag/ZnO (100 nm)/P Ag/ZnO (90 nm)/Pt | | amorphous ZnO | [88,98] | $-0.2$ | 0.24 | $>10^7$ |
| | | | | $-0.6$ | 0.5 | $10^2$ |
| Ti/ZnO (90 nm)/Pt | | | [98] | $-0.9$ | 1.1 | 10 |
| Ag/ZnO (70 nm)/Graphene | | | [77,90] | $-3.11$ | 3.81 | 30 |
| Cu/ZnO (23 nm)/ITO | | no O treatment | | no switching behavior | no switching behavior | no switching behavior |
| Ag/ZnO (70 nm)/Graphene | | O treatment | [77] | ND | ND | $\sim 10$ |

## 4. Modulation Mechanisms and Utilizing ZnO and 1D/2D Materials

There are many factors influencing RRAM devices, including the choice, morphology, and deposition of the active layer and electrodes, which have been discussed previously. Another way to improve the performance is to modify the layer via illumination or other treatment, doping of ZnO layers, or implementing 2D and 1D nanostructures.

### 4.1. Influencing Pure ZnO Monolayer

It is possible to modify the ZnO RRAM devices' properties using UV light. Illumination is an efficient way to break Zn–O bonds, generating oxygen ions and radicals. After illumination, the current conduction mechanism of the ZnO RRAM device was changed from Schottky emission to Poole–Frenkel conduction [84]. UV illumination can be used to better RS performances in terms of the device's endurance and current values [104]. Another procedure improving ZnO films' properties is neutral O beam treatment [90]. It allows for creating RRAM devices with very thin ZnO films (23 nm), which otherwise show no switching behavior as a result of a large leakage current. The treatment is effective in decreasing O vacancy defects in the sputtered-ZnO film and promotes the formation of the conducting filament at a lower operation current.

Saini et al. [105] performed a study on the influence of illumination of specific wavelength light in conjunction with applied voltage. It has been reported that illumination modulates the bipolar switching behavior. The $V_{set}/V_{reset}$ for RS obtained for different wavelengths (in the range of 300–700 nm) appear in the middle of the $V_{set}/V_{reset}$ obtained under dark and white light illumination. Light wavelength adds up an extra control parameter in conventional memristor devices [105].

Many studies researched the influence of doping on RRAM device parameters. The incorporation of dilute concentration of dopant having a valence state different from that of the host cation enables controlled incorporation of vacancy defects [106]. As can be shown in XPS spectra, certain dopants result in increasing oxygen vacancies [107,108] attributed to CF formation mechanism. Research conducted on ZnO films confirms that the forming energy of oxygen vacancies lowers after doping with typical p-type impurities [109]. Additionally, first-principle calculations performed for Ti-doped and Co-doped ZnO reveal that the formation energies of oxygen vacancies had the minimum value when it was located at the

next-nearest neighbor to Ti atoms [110,111]. In the following paragraphs, the most popular dopants will be discussed, but it is worth mentioning the transition metals such as Fe [112], Cr [107], Ti [110,113], Ag [114,115], which are not so common. Among the transition metals, Ti, Ag, and Hf [116] also appear in electrodes or additional layers which will be discussed in the next point. Other frequent dopants include Pr [103,117], N [85,118], Mg [119,120], Ge [121,122].

In Li-doped ZnO-based RRAM devices [123–125], Li can go to the interstitial and substitutional site in the lattice structure of ZnO, leading to improved crystallinity [80]. Compared with the pure ZnO-based device, the proposed ZnO:Li devices achieve better bipolar resistive switching characteristics, including a high ON/OFF current ratio, a low set voltage (<1.0 V), and reset voltage [125]. Characteristic two-, three-, or four-step RESET behavior of the LZO RRAM devices can be attributed to the effect of Li addition. The type of multistep behavior, however, is controlled by tuning the compliance current [118]. The performance of the device is dependent on the concentration of dopant, which has been researched by Zhao et al. It has been shown that in the Li-doped ZnO, increasing Li content results in a decrease in oxygen vacancies [125,126].

Cu doping [105,127–130] allows for investigating the PR behavior (polarization rotation, also called polarization switching or polarization orientation) of structures. With the increasing copper concentration, HRS and LRS are more distinguished. Doping copper allows for defect engineering and increased charge storage and polarization rotation behavior [127]. The switching mechanism in Cu:ZnO is believed to be due to the formation of oxygen vacancies, not because of filamentary formation, as the Cu incorporated into the ZnO lattice does not bond with oxygen; rather, it creates oxygen vacancies and gets synched with them [128]. The direct bandgap of Cu:ZnO is lower than that of the conventional ZnO. Lowered bandgap implies that the concentration of Cu impurities creates localized states, which are defects caused by the unsaturated bonds [130]. Higher Cu content can shift the $V_{set}$ and $V_{reset}$ values [109].

In Mn-doped ZnO-based RRAM devices [131–134], the incorporation of Mn can lead to a stable bipolar resistive switch (BRS). $V_O$ defects, which are more prominent in Mn-doped ZnO, are the primary determinant of BRS in ZnO because $V_O$ defects migrate easily in binary oxides under an electric field, generating $V_O$ -based conductive filaments [134]. The ZMO device also showed high endurance characteristics [133]. Another study on ZMO suggests that BRS requires additionally a suitable bottom electrode [131].

Co-doped ZnO [108,135,136] is a seminal spin-tronic material that shows strong ferromagnetism under both insulating and metallic states. RRAM using Co:ZnO as active layer shows stable RS during repeated sweep cycles and stable bipolar RS characteristics, and exhibits magnetic modulation with the alternation of set and reset processes, which allows for obtaining four logic states [135]. Additionally, pure ZnO has a much smaller memory window compared with Co-doped (2 at% and 5 at%) [136]. Co dopant can be used to control the defect concentration in ZnO films, mainly oxygen vacancies, which further improves RS performance [108].

In Al-doped ZnO-based RRAM [137–139], multiple resistance states could be obtained. The modulation is performed by controlling the stop voltage (five HRS and one LRS) and setting the ICC (three LRS and one HRS) [139]. Al nanoparticles have been used to lower the potential barrier of the active layer and electrode's interface [140]. Al nanoparticles as the tip electrode can effectively enhance the local electric field; therefore, when a positive voltage is applied to the device, the CFs will first form near the Al nanoparticles due to the local electric field effect [140]. Doping of Al modulates the oxygen vacancy concentration in ZnO matrix as aluminum ion can substitute for the zinc ion. The cationic imbalance in ZnO matrix induces defects. At low voltage, $Al^{3+}$ can act as a chemical anchors for vacancies through which the $O^{2-}$ can move [141]. The increased oxygen deficiency increases the conductivity of the ZnO film [142]; therefore, the concentration of Al in the ZnO film can modulate the device performance, as higher concentration (20 at. %) leads to higher LRS/HRS ratio. On the other hand, excessive amounts of Al incorporated into ZnO causes

instabilities in the device retention and endurance, as $Al_2O_3$ clusters may act as carrier traps [143].

Ammonia treatment is a process of introducing $NH_4$ ionic bonds into ZnO films. Ammoniated ZnO devices retain the characteristics of ZnO devices but have improved reliability. Additionally, the power consumption of RRAMs is demonstrated to be reduced by 80%. The change is attributed to additional -$NH_x$ functional groups repairing the Zn- and O- dangling bonds [144].

Another way to modulate RRAM parameters is to control the thickness of the active layer [94,119]. The Li-doped ZnO-based RRAM with thicker ZnO layers shows higher ON/OFF current ratio $I_{ON}/I_{OFF}$ and the set voltage $V_{set}$ [119]. Additionally, another study performed on sol-gel ZnO-based RRAM devices demonstrates the relation between forming voltage and the thickness of the ZnO layer [94]. The study on oxygen treatment mentioned above attributes the lack of BS behavior to a large leakage current which is also connected with active layer's thickness [90].

*4.2. 2D Materials*

Introducing additional layers can improve the RRAM performance. One of the techniques is creating bilayers (BL) by adding a layer of the electrode's metal oxide [145], or adding a doped ZnO layer [119,132,146,147] or an additional layer between ZnO and the electrode [119]. Trilayers may be created by inserting an additional layer between two ZnO layers [148] or before and after the ZnO layer [149].

One of the ways of constructing bilayers for RRAM devices is taking one material that lacks oxygen vacancies and another that has an abundance of them. Therefore, the latter can be used as the oxygen vacancy reservoir, attracting the oxygen out of prior, as in $Ga_2O_3$/ZnO -based devices [150].

When Cu/ZnO/AZO and Cu/CuO/ZnO/AZO devices were compared [145], it was noted that the resistive switching characteristic was significantly better in BL structures. There, the CuO layer is a "reservoir" of oxygen ions in the set process and acts as an oxygen ion "supplier" in the reset process, which plays a critical role in recovery/rupture of filament paths. Similarly, a Ga-doped ZnO nanorod layer as in ITO/(GZO)/ZnO/ITO device [146] acts as the oxygen reservoir for high-performance RS behavior. The HRS/LRS ratio is increased and the distribution of each state is very narrow, compared to pure ZnO devices.

Studies by Simanjuntag et al. [151,152] propose the addition of a high-resistivity $ZnO_2$ layer to lower the operation current needed for the formation of a conducting bridge. Similarly, degradation of operating current has been observed by Wu et al. [153]. In this study, the addition of highly resistive $HfO_x$ thin film thickness could modulate the barrier at the $HfO_x$ interface. Later, the performance improvement was explained in the study performing first-principles calculation [154]. There, a middle resistance state (MRS) speculation was proposed in which a conductive filament was formed by Ag atom and oxygen vacancy at the same time and was responsible for degradation of HRS. The introduction of $HfO_2$ layer modulates the conductive channel of oxygen vacancy, which improves the device performance.

High current for the reset process means high power dissipation. Introducing an additional layer inside the ZnO active layer, as in the TiN/ZnO/Ni/ZnO/Pt device [38], reduces the reset current. Another effect is increased forming voltage, which is explained by the behavior on the interface between ZnO and Ni. Ni diffuses into the ZnO thin film, which causes impurity energy level and defect when sputtering on the ZnO film. The opposite situation, in which the ZnO layer is surrounded from both sites by inserting layers, has also been constructed as in the TiN/$Al_2O_3$/ZnO/$Al_2O_3$/TiN device [149]. The motivation for this structure is the fact that layers on both sides of ZnO would help stabilize the local oxygen migrations for the formation and rupture of the CF during the continuous switching cycles. The modification results in increasing memory switching characteristics such as high HRS/LRS ratio, endurance, and stable retention at high temperatures.

The addition of conventional 2D materials may also change the properties of the ZnO layer itself. Shen et al. [155] created a transistor with resistance-switching properties using $MoS_2$ and ZnO. $MoS_2$ and ZnO have been shown to possess hysteretic characteristics, which are attributed to the adsorption/release of $O_2$ and water molecules on the surface of the channel and electron capture at the 2D material/oxide interface. The $MoS_2$/ZnO heterojunctions showed an ON/OFF ratio of $10^4$ and significant rectifying behavior with a forward-to-reverse bias current ratio. Kadhim et al. [156] obtained the broad spectrum with self-colored ZnO layers on the Ti foil, varying the sputtering time of $MoS_2$. The structures exhibited different coloration and resistive switching responses due to the thickness of the $MoS_2$ layer. In the work of Jagannadham [157], two-dimensional semiconductor $MoS_2$ films in combination with ZnO are used to form the diode–memristor structure. The height barrier for Schottky conduction in the switching from LRS to HRS is higher for positive polarity and reduced for a negative polarity by forming a p-n junction with a memristor.

Numerous authors used graphene and reduced graphene oxide (rGO) to enhance ZnO active layer properties. Zhou et al. [158] proposed the hybrid of zinc oxide nanorods and a reduced graphene oxide hybrid nanostructure to create flexible RRAM with an ON/OFF ratio of $10^7$. Khanal et al. [159] proposed an active layer of the ZnO–rGO composite for analyzing the synaptic behavior of a structure. Cardarilli et al. investigated a sol-gel ZnO-graphene oxide layer, two-terminal MIM with Al and FTO electrodes [160]. In Khanal et al.'s [161] work, a thin film was sandwiched between Ag and FTO electrodes and annealed at 500 °C to create oxygen vacancies. The memristor exhibited a ratio $R_{on}/R_{off}$ of approximately $10^3$. Aziz [162] and Izam [163] investigated ZnO–graphene hybrids. In the work of Aziz, ZnO was grown on a glass substrate using thermal chemical vapor deposition at different substrate temperatures of 350 °C, 450 °C, and 550 °C, and graphene in water solution was transformed into a thin film using a water bath at 90 °C. The material was sandwiched between Pt electrodes. The ZnO–graphene devices exhibited bipolar resistive switching characteristics with a slightly smaller memristive window than those without graphene. The addition of graphene upgraded the transitioning cycle stability. Izam and colleagues investigated the influence of dip-coating speed on the memristive properties of the ZnO-decorated graphene film.

### 4.3. 1D Materials

Various ZnO low dimensional structures have been used as active layers, such as nanorods [96,146,164–166], nanowires [101], and nanoislands [167]. The resistive switching mechanism in ZnO nanorod layers has been described by Chang et al. [168]. Oxygen vacancies and/or Zn interstitials could easily condense to form tiny filaments at the surface of the single crystalline ZnO nanorods because the mobility of defects at the surfaces is much higher than that in the single crystal. The gathering of these tiny filaments at the surfaces of the individual ZnO nanorods causes the formation of straight and extensible conducting filaments along the direction of each nanorod. Cathodoluminescence imaging spectroscopy performed in Tseng et al. [169] revealed the $V_o$-induced green emission distributed on the nanorod sidewall. The oxygen vacancy, with single positive charge ($V_o$), has a deep donor level that can trap an electron. The following transition results in neutral oxygen vacancies $V_o$, which are conductive shallow donors. In ITO/ZnO NRL/Al devices, the switching resistance mechanism can be explained in the following manner: when the applied voltage reaches above the SET voltage, the injected electrons fully occupy the trap level, and subsequent electrons can transport to the ITO without being trapped. Under negative bias, the captured electrons are released and attracted to the Al electrode due to the negative electric field which results in an increased resistance. The conductive filaments are bridged on the nanorod sidewall. Nanorods ensure higher stability via more stable straight filaments than branched ones [168]. Similarly, in ZnO nanoisland devices, conducting filaments are formed on the edges of the ZnO nanoisland [167].

## 5. Conclusions

Resistive random access memory stands out among memory technologies due to its scalability, high-speed operation, and low power consumption. Zinc oxide is one of the most promising candidates for RRAM active layers. The combination of suitable bandgap, thermal conductivity, chemical stability, and above all, flexibility, transparency, and biocompatibility, allow for a wide range of designed devices. Extensive studies have been dedicated to finding the influence of the thin film deposition methods on device performance. It has been found that even cheap chemical routes allow for a high HRS/LRS ratio. The selection of electrode, its size, and interface with the active layer have tremendous effects on conduction mechanisms. To further alter the characteristics, rapid thermal annealing, UV, and visible light illumination can be used. Doping of ZnO layers not only can enhance or stabilize the performance but also introduce multistep or polarization rotation behavior. Bi- and trilayers are used to, among other uses, modify the interface barrier and control the oxygen migrations. The incorporation of low dimensional structures such as nanowires, nanobelts, and nanoislands creates a path for stable straight filaments. This work may give a valuable insight into the development of ZnO-based RRAM technology and its prospects.

**Author Contributions:** Conceptualization, E.N.; methodology E.N. and E.C.; software, E.N.; validation, E.N., E.C. and M.S.; formal analysis, E.N. and E.C.; investigation, E.N. and E.C.; resources, E.N., E.C. and M.S.; data curation, E.N., E.C. and M.S.; writing—original draft preparation, E.N. and E.C.; writing—review and editing, E.N., E.C. and M.S.; visualization, E.N.; supervision, M.S.; project administration, M.S.; funding acquisition, M.S. All authors have read and agreed to the published version of the manuscript.

**Funding:** This research was funded by the Ministry of Science and Higher Education, grant number 0511/SBAD/2251.

**Data Availability Statement:** This paper consists of review material; thus, there were no new data used for preparation.

**Conflicts of Interest:** The authors declare no conflict of interest.

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
