# Peer review of "ZnO and ZnO-Based Materials as Active Layer in Resistive Random-Access Memory (RRAM)"

_crystals, doi:10.3390/cryst13030416_

Round 1

Reviewer 1 Report

The paper “”ZnO and ZnO based materials as active layer in RRAM”” explains what is RRAM, gives its history and considers different materials as active RRAM media.

The paper is well prepared, written in a good language, clear and yet highly professional.

papers given as references are relevant, novel and appropriately cited.

I only mentioned one flaw. The paper contains parts of the original template in sections Data availability statement, Acknowledgements, Appendixes. They should be either filled or deleted.

Author Response

Thank you very much for your really good review. We have improved the sections due to your suggestions.

Reviewer 2 Report

Here, Nowak et al have written a review paper on ZnO materials and on their use as RRAM active layer. An analytic overview on the influence of different dopants on the characteristics of the device is presented.

It is a very well-organized review paper.

However, in the abstract part, it is written: “the inclusion of inserting layers and nanostructures into the ZnO-based RRAM has been investigated”, but there is not any new research investigation in the mns.

The authors should clarify if there is any new research data.

If there are new data, they should provide them explicitly.
If there are not any new data, they should rephrase the abstract.  

Author Response

Thank you very much for the suggestion of improving the abstract. As you have mentioned, the manuscript did not contained additional data. Thus, the appropriate corrections have been included in the abstract.

Reviewer 3 Report

This review manuscript discusses the recent advances in ZnO-based Resistive random-access memory devices. The review starts by presenting the fundamentals of resistive memory including polarity-dependent and -independent mechanisms, correlating them with various common materials used for RRAM devices. Given the technological significance of RRAM and the ubiquity of ZnO as a functional material, I believe this review paper can be beneficial to Crystal’s readers, subject to the following minor revisions:

On page 13, when discussing the performance of Al and transition metal doping, the lowering of oxygen vacancy formation energy as a result of doping should be explicitly discussed and cited, as stipulated in the following references among others:

Y B Zhang et al. “Structural and magnetic stability of dopants in ZnO-based dilute magnetic semiconductor” J. Phys.: Condens. Matter 23 066004; 10.1088/0953-8984/23/6/066004

Janotti and Van de Walle “Fundamentals of zinc oxide as a semiconductor” Rep. Prog. Phys. 72 (2009) 126501; 10.1088/0034-4885/72/12/126501

RRAM must be defined in the abstract and the title. CMOS and CBRAM are not defined at the first instance either.

The article needs thorough language and style polishing. VO --> VO

Author Response

Thank you very much for all of the suggestions. Those are our replies to your remarks.

The lowering of oxygen vacancy formation energy in the case of transition metal doped ZnO is discoussed in more details.

Abbreviations have been added and the language has been improved

Round 2

Reviewer 2 Report

The mns has been improved and some uncertainties have been clarified.

Therefore, I recommended at the present form.